# The Effect of Marine *n*-3 Polyunsaturated Fatty Acids on Heart Rate Variability in Renal Transplant Recipients: A Randomized Controlled Trial

**DOI:** 10.3390/nu11122847

**Published:** 2019-11-20

**Authors:** Hanne Sether Lilleberg, Simon Lebech Cichosz, My Svensson, Jeppe Hagstrup Christensen, Jesper Fleischer, Ivar Eide, Trond Jenssen

**Affiliations:** 1Department of Nephrology, Akershus University Hospital, 1478 Lørenskog, Norway; m.h.s.svensson@medisin.uio.no (M.S.); ivar.anders.eide@gmail.no (I.E.); 2Department of Health Science and Technology, Aalborg University, 9220 Aalborg, Denmark; 3Department of Clinical Medicine, University of Oslo, 0316 Oslo, Norway; t.g.jenssen@medisin.uio.no; 4Department of Nephrology, Aalborg University Hospital, 9000 Aalborg, Denmark; jeppe.hagstrup.christensen@rn.dk; 5Steno Diabetes Center Aarhus and Zealand, 8000 Aarhus, Denmark; jesper.fleischer@clin.au.dk; 6Department of Organ transplantation, University hospital of Oslo, 0372 Rikshospitalet, Norway

**Keywords:** marine *n*-3 PUFA, heart rate variability, resting heart rate, renal transplant recipients

## Abstract

Resting heart rate (rHR) and heart rate variability (HRV) are non-invasive measurements that predict the risk of sudden cardiac death (SCD). Marine *n*-3 polyunsaturated fatty acid (PUFA) supplementation may decrease rHR, increase HRV, and reduce the risk of SCD. To date, no studies have investigated the effect of marine *n*-3 PUFA on HRV in renal transplant recipients. In a randomized controlled trial, 132 renal transplant recipients were randomized to receive either three 1 g capsules of marine *n*-3 PUFA, each containing 460 mg/g EPA and 380 mg/g DHA, or control (olive oil) for 44 weeks. HRV was calculated in the time and frequency domains during a conventional cardiovascular reflex test (response to standing, deep breathing, and Valsalva maneuver) and during 2 min of resting in the supine position. There was no significant effect of marine *n*-3 PUFA supplementation on time-domain HRV compared with controls. rHR decreased 3.1 bpm (± 13.1) for patients receiving marine *n*-3 PUFA compared to 0.8 (± 11.0) in controls (*p* = 0.28). In the frequency domain HRV analyses, there was a significant change in response to standing in both high and low frequency measures, 2.9 (*p* = 0.04, 95% CI (1.1;8)) and 2.7 (*p* = 0.04, 95% CI (1.1;6.5)), respectively. In conclusion, 44 weeks of supplemental marine *n*-3 PUFAs in renal transplant recipients significantly improved the cardiac autonomic function, assessed by measuring HRV during conventional cardiovascular reflex tests.

## 1. Introduction

Sudden cardiac death (SCD) is a major cause of death in patients with chronic kidney disease (CKD) [1,2]. Heart rate variability (HRV) and resting heart rate (rHR) are non-invasive measurements that may predict SCD risk [3]. HRV can be measured by the beat-to-beat variability in heart rate with either time-domain or frequency-domain analyses during passive or controlled active testing. Autonomic dysfunction, reflected in an imbalance between sympathetic and parasympathetic activity, may increase rHR and decrease HRV [4,5]. Previous studies have shown that patients with ischemic heart disease [5], diabetes mellitus [6], and chronic kidney disease (CKD) [4,7,8,9], with a well-documented high risk of SCD, have an attenuated HRV often in combination with a high rHR. After renal transplantation, HRV increases due to improved autonomic function [10,11], but the risk of SCD after renal transplantation still remains high [1,12].

Fish and seafood are the main sources of marine *n*-3 polyunsaturated fatty acids (PUFAs), including the two major PUFAs, eicosapentaenoic acid (EPA 20:5) and docosahexaenoic acid (DHA 22:6) [13]. A regular intake of dietary or supplemental marine *n*-3 PUFA has shown a positive effect on autonomic cardiac function, with a decrease in rHR and an increase in HRV in several patient groups, suggesting it might protective against SCD [3,14,15]. This effect seems to be dose-dependent [16,17], and might translate into a specific threshold level for lowering the risk of all-cause mortality and SCD [13,18].

Two recent observational studies investigated the relationship between the plasma content of marine *n*-3 PUFA and risk of SCD in renal transplant recipients, showing that high levels of plasma marine *n*-3 PUFA were inversely associated with all-cause mortality and SCD [12,19]. A number of small randomized controlled trials have shown that supplementation with marine *n*-3 PUFA may have a beneficial effect on lipids and blood pressure in renal transplant recipients [20], but to date, no large intervention studies with cardiovascular endpoints exists. The present study is a sub-study of the recently published trial “The effect of Omega-3 Fatty Acids in Renal Transplantation” (ORENTRA) [21] and, to our knowledge, was the first study to investigate the effect of marine *n*-3 PUFA on HRV and rHR in renal transplant recipients.

The aim of this study was to examine the effect of high dose marine *n*-3 PUFA on rHR and time- and frequency-domain HRV in renal transplant recipients.

## 2. Material and Methods

### 2.1. Study Design

The study was a single-center, randomized, double-blind, controlled, parallel-grouped trial in Norwegian renal transplant recipients. Eligible patients were adults aged above 18 and had a functional kidney graft with an eGFR > 30 mL/min/1.73 m^2^ at 6–8 weeks after the renal transplantation and provided written informed consent. Exclusion criteria were allergy to the study drug, fish, or other seafood, patients receiving a kidney from a donor older than 75 years, participation in other clinical trials, and citizenship in a foreign country.

Patients were randomized to either marine *n*-3 PUFA (460 mg/g EPA and 380 mg/g DHA (Omacor)) given as one capsule of 1 g three times daily, or three capsules of 1 g extra virgin olive oil, per day. Omacor and control oil in capsules were both provided by Pronova Biopharma. Randomization codes were generated by a computer. Study participants, care providers, and investigators were kept blinded during the study period. 

The primary endpoint of the main trial was mGFR and power calculation was based on an absolute difference in mGFR of 8.0 mL/min/1.73 m^2^, which required 132 patients to be enrolled. This sub-study investigated a secondary endpoint, which was absolute change in HRV during follow up. Post hoc additional analyses were done in a subgroup of patients with baseline plasma marine *n*-3 PUFA < 6 wt. %. Further details on study design, participants, and sample size have previously been published [21].

### 2.2. Measurements

Measurements were performed 8-week post-transplant (baseline) and one year after transplantation (end of study). HRV and rHR measurements were obtained in the morning, followed by blood samples, including aliquots used for fatty acid analysis. Marine *n*-3 PUFA was defined as the sum of EPA and DHA quantified as weight in percentage (wt. %) of total plasma phospholipid fatty acid. Gas chromatography was used to determine fatty acid distribution according to standard procedures [22].

HRV was calculated in time and frequency domain during conventional cardiovascular reflex tests (response to standing, deep breathing, and Valsalva maneuver) and during 2 min of resting in the supine position. The tests were performed by laboratory technicians using a Vagus^TM^ (Medicus Engineering, Aarhus, Denmark) [23]. This device automatically records an ECG signal with a sampling frequency of 1000 Hz, from which HR and HRV are deduced and shown on a display. Three cardiovascular reflex tests were performed using a standard protocol and recommendations, including (1) response to standing, (2) deep breathing, and (3) Valsalva test [24]. The test of response to standing is also known as the 30/15 ratio because tests indicate that the shortest R–R interval (time between beat in milliseconds) is found around 15th beat and the longest R–R interval is found around the 30th beat. The ratio is obtained by continuously measuring R–R intervals from when an individual is in a supine position to about a minute after the patient stands up. The 30/15 ratio is largely a measure of parasympathetic function. The deep breathing test is a test where the device acts as a pacer for the patients breathing. The pace of the breathing is six breaths per minute and the device paces the patient through the instructions on the display. During deep inspiration the heart rate increases, and during expiration the heart rate decreases. After 60 s of deep breathing exercise, the mean ratio of heart rates is calculated. The Valsalva test is a response test where the patient exhales with a pressure of 40 mmHg for 15 s. The Valsalva ratio is calculated as the longest and shortest R–R intervals during forced expiration in 15 s against a fixed resistance of 40 mmHg and 45 s of normal breathing. The Valsalva ratio is a measure of both parasympathetic, sympathetic function, and baroreceptor function. Each test was followed by a 1 min break. The cardiovascular reflex tests were performed by trained examiners and were preceded by a 30 min period of rest in a supine position in a quiet and isolated room. Antiarrhythmic medication, including betablockers, were not taken the morning before HRV measures.

The following HRV matrices were used: In the time domain we used standard deviation of normal-to-normal intervals (SDNN). The HRV parameter SDNN is a measure of combined sympathetic and parasympathetic activity. In the frequency domain, we used the low-frequency (LF) 0.04–0.15 Hz and the high-frequency 0.15–0.4 Hz components. The LF components are influenced by sympathetic, parasympathetic, and baroreflex sensitivity. The high frequency (HF) band from 0.15–0.4 Hz is influenced by parasympathetic and the normal breathing rhythm, mainly contributing to power with a center frequency around 0.3 Hz [25].

### 2.3. Ethics

The trial was approved by the Regional Committees for Medical and Health Research Ethics in Norway and The Norwegian Medicines Agency. The trial was completed in accordance with the Declaration of Helsinki. Clinical. Trials. Gov. identifier NCT 01744067.

### 2.4. Statistical Analyses

Primary statistical analyses were performed on an intention-to-treat (ITT) basis and findings reported accordingly to CONSORT guidelines [26]. In addition, a secondary per-protocol (PP) analysis was performed. Baseline data, follow-up data, and effect estimates with precisions have all been reported accordingly to the guidelines [26]. A GLM model (type: general linear model) was used to assess rHR and HRV effect estimates from baseline to follow-up in an unadjusted analysis. This type of method is preferred as it takes the potential of unbalanced groups and risk of regression towards the mean into consideration [26]. Missing baseline data were median-imputed [27]. Missing data at follow-up were assumed missing at random. Imputation models included all baseline variables (Table 1). Linear regression was used for continuous variables and logistic regression for categorical variables. Frequency domain HRV measures were log-transformed for analysis. The primary analysis was based on the imputed data. Complete case analyses (PP) were also included to check sensitivity of the main trial findings. Moreover, a subgroup analysis of patients with plasma marine *n*-3 PUFA < 6 wt. % was assessed to evaluate whether the impact of intervention would be more evident in patients with a lower baseline plasma marine *n*-3 PUFA. A two-sided *p* value < 0.05 was considered significant. SPSS version 24.0 was used for statistical analyses.

## 3. Results

Between 15 June 2013 and 15 June 2014, 298 patients received a kidney transplant at Rikshospitalet in Norway; 132 were included in the study. Baseline characteristics are presented in Table 1. Complete baseline data have previously been published [21]. During the study, plasma levels of marine *n*-3 PUFA increased significantly in the intervention group, compared to no change in the control group, as previously published [21].

### 3.1. Time and Frequency Domain HRV

The HRV measurements at baseline and follow up are presented in Table 2. Four patients had documented arrhythmia at baseline. Time-domain HRV data were comparable in the two groups at baseline, and HRV increased slightly in both groups during the study period (Table 2). There was a decrease in rHR in the marine *n*-3 PUFA group of 3.1 bpm (± 13.1), compared to 0.8 (±11.0) in the control group (Table 2). Both groups had an increase in SDNN, 7.8 (± 23.3) in the marine *n*-3 PUFA group compared to 3.4 (± 21.8) in the control group. No difference between groups was found in heart rate response to the active tests; orthostatic, E:I, or Valsalva. In frequency-domain HRV, the orthostatic ratio analysis was significantly different at baseline. The control group had a significantly higher level of both low and high frequency HRV, 28.1 and 7.1, compared to 14.6 and 3.0 in the marine *n*-3 PUFA group.

The intervention effects of marine *n*-3 PUFA compared to the control group for both ITT (*n* = 132) and PP (*n* = 102) population analyses are presented in Table 3. There were no significant differences between groups in rHR or SDNN during supine resting. In the time domain, Valsalva ratio was the only parameter with a significant reduction of 0.1 ms^2^ (*p* = 0.04, 95% CI (−0.2; −0.01)) in the ITT population. In the PP population, the orthostatic test increased significantly by 0.2 ms^2^ (*p* = 0.04, 95% CI (0.01; 0.4)) in the marine *n*-3 PUFA group compared to the control group. In the frequency domain, the intervention effect was significant in the orthostatic test in both low and high frequency in the ITT and PP population. The effect of marine *n*-3 PUFA was 2.9 (*p* = 0.04, 95% CI (1.1; 8)) and 2.7 (*p* = 0.04, 95% CI (1.1; 6.5)) for respectively HF and LF domains in the ITT population. There was no significant effect in E:I test or Valsalva test between groups.

### 3.2. Subgroup Analysis

Subgroup analysis was performed in the 63 patients with plasma marine *n*-3 PUFA < 6 wt. %. In the marine *n*-3 PUFA group, there was a mean reduction in rHR of 5.0 ± 2.8 bpm, but this was still not significant when compared to the control group (data not shown). No significant differences were found between the two groups in the time-domain or frequency-domain HRV (data not shown).

## 4. Discussion

The main findings of this study were that high doses of marine *n*-3 PUFA supplementation in renal transplant recipients for 44 weeks significantly improved frequency-domain HRV in response to standing, but did not affect time-domain HRV or rHR.

Previous studies have shown improvement in HRV in end-stage kidney disease patients after receiving a renal transplant [10]. Cashion et al. conducted a study of 90 kidney- and 30 kidney-pancreas transplant recipients, where patients without diabetes receiving a kidney transplant had an improvement in HRV after 6 months, as did patients with diabetes receiving a kidney-pancreas transplant after 12 months [11]. This was confirmed in a study investigating 24 h time- and frequency-domain HRV prior to and at 6 months after kidney and kidney–pancreas transplantation, with an increase in HRV in all 57 patients [28]. Our results showed an increase in HRV in both groups in the study period from baseline, which was 8-week post-transplant, to 44-week post-transplant, with an additional improvement from intervention on frequency domain parameters (response to standing). Thus, it seems that renal transplantation improves autonomic function and HRV per se, and our data suggest that marine *n*-3 PUFA supplementation might have a further beneficial effect on autonomic function in this setting.

However, only the frequency-domain HRV in response to standing was significantly improved after supplementation with marine *n*-3 PUFA. The orthostatic test is the result of a transient decrease in blood pressure and increase in the HR following translocation of blood due to active standing from the supine position. Failure to increase HRV component LF on standing reflects an impaired sympathetic response or depressed baroreflex sensitivity [29,30]. In addition, HRV in response to standing might be more precise in patients with severe autonomic neuropathy, with a better HRV amplitude, making it possible to differentiate between an actual signal and background noise [25]. On the other hand, frequency-domain analyses may be distorted by verbal activity and respiratory changes, making the results less reliable [31]. 

In general, studies of the effect of marine *n*-3 PUFA on HRV have shown diverging results. A meta-analysis from 2013, including studies of patients with different co-morbidities, concluded that there was an overall effect on HRV only on parameters reflecting an increased vagal tone (frequency domain) [32]. To our knowledge, no previous studies have examined the effect of marine *n*-3 PUFA consumption on HRV in renal transplant recipients, and only a few studies exist in patients with CKD. Previously, two small RCTs in HD patients did not find an effect of supplementation with marine *n*-3 PUFA on HRV. [8,33]. In the largest study to date, Rantanen et al. randomized 112 HD patients to 2 g of marine *n*-3 PUFA or control for 12 weeks and showed an improvement in HRV parameters reflecting vagal tone and a significant reduction in rHR, but no significant effect on the primary endpoint, SDNN [34]. There are, however, multiple factors that may affect HRV in patients with CKD, including frequent use of medications. In our study, approximately 40% of the patients were treated with anti-arrhythmic medication, which may have affected the results.

In our study, baseline levels of marine *n*-3 PUFA were relatively high. The Norwegian population in general has a documented high average intake of fish, equivalent to 2–3 meals per week [35], and concordantly plasma marine *n*-3 PUFA levels in the Norwegian population are high compared to levels of marine *n*-3 PUFA in other populations [36]. Previous data in Norwegian renal transplant recipients [12], and high baseline levels over 6 wt. % in the present study, confirm this. 

It has previously been suggested that specific threshold levels might exist for the protective effects of marine *n*-3 PUFA, based on pooled data from epidemiological studies showing an inverse association with a moderate intake (two servings/week) and all-cause mortality [13]. The risk for all-cause mortality was not lower with an intake above two servings per week, suggesting a possible threshold effect. Recent meta-analysis of RCTs also confirmed a dose–response relationship [37]. Additionally, different effects of marine *n*-3 PUFA have different threshold levels. The anti-arrhythmic and antihypertensive effect may be reached with typical dietary doses [18,38], whereas achieving, for example, a triglyceride-lowering effect would require high-dose supplementation [39].

Finally, previous data also suggest that dietary consumption or supplementation with marine *n*-3 PUFA may be most effective for risk reduction in populations with low baseline levels [40]. An omega-3 index has also been used as a biomarker to predict risk of cardiovascular (CV) death [41]. This index defines low levels as <4 wt. % and high levels as >8 wt. %, corresponding to high vs. a low risk for CV death. We performed an additional subgroup analysis of patients with a baseline level of marine *n*-3 PUFA below 6 wt. %, which did not alter the results. As the mean dietary intake of marine *n*-3 PUFA in our study was high, very few patients had levels below the suggested threshold level of 4%. The small sample size of our study did not allow us to explore this further. 

In a large cross-sectional study of 5096 men and women, dietary intake of fish was associated with lower rHR [42]. In line with other data, their results also supported a threshold effect from marine *n*-3 PUFA at a dietary consumption of approximately 300 mg/day. In renal transplant recipients, one large cross-sectional study showed that a high level of marine *n*-3 PUFA was associated with lower rHR 10 weeks after transplantation [43]. A previous meta-analysis of marine *n*-3 PUFAs’ effect on rHR in randomized trials found that marine *n*-3 PUFA decreased rHR by 1.6 bpm, and by 2.5 bpm in trials with baseline rHR above 69 [44]. Another meta-analysis showed a significantly reduced rHR after supplementation with marine *n*-3 PUFA, with a mean reduction of 2.2 bpm overall and a more pronounced effect from DHA than EPA [45]. In a recent RCT of patients treated with chronic HD, a similar reduction in rHR (2.5 bpm) was shown after treatment with 2 g of marine *n*-3 PUFA for 12 weeks [34]. 

In our study, the patients had a relatively high mean rHR at baseline, 74.4 bpm in the marine *n*-3 PUFA group and 73.0 bpm in the control group, with a reduction in rHR in both groups during the study period. Although our results showed a larger reduction in rHR in marine *n*-3 PUFA compared with controls (3.1 vs. 0.8 bpm), in line with previous studies, the difference was not significant. As previous data regarding marine *n*-3 PUFA and rHR are relatively consistent, our results might have been due to the small sample size of our study. 

Two recent observational studies in renal transplant recipients have shown that high levels of marine *n*-3 PUFA are associated with better patient survival [12,19]. In a large cohort study, Eide et al. followed 1990 Norwegian renal transplant recipients for a median of 7 years and found that high levels of marine *n*-3 PUFA at baseline were independently associated with lower risk of all-cause mortality, and in particular a lower risk of SCD [12]. Using the lowest quartile of marine *n*-3 PUFA as a reference, the hazard ratio for SCD in the highest quartile was 0.08 (95% CI 0.02–0.35). Neto et al. showed similar results in an observational study with 627 renal transplant recipients [19], with a lower risk of all-cause and cardiovascular mortality in patients with high levels of marine *n*-3 PUFA. These studies are in line with epidemiological data both from the general population [13], and patients with CKD [46,47]. 

As for data from intervention studies, there has been ongoing controversy regarding the effect of marine *n*-3 PUFA on CV events and death from cardiac causes. However, in recent years, several new studies have been published, and the conclusion from two updated meta-analyses was that supplementation with marine *n*-3 PUFA reduces both CV events and CV death [37,48]. In patients with CKD, one previous RCT in patients treated with HD showed that supplementation with 1.7 g of marine *n*-3 PUFA significantly reduced the number of myocardial infarctions with no effect on all-cause mortality [49]. In renal transplant recipients, no intervention studies with marine *n*-3 PUFA on cardiovascular endpoints exist. However, such a study would be warranted considering the low levels of marine *n*-3 PUFA in patients with CKD in general, the high risk of CV events, including SCD, and recent epidemiological data in renal transplant recipients. In our study of rHR and HRV as surrogate markers for SCD, the results might have been affected by a high background intake of marine *n*-3 PUFA above a certain threshold level, and baseline levels should be taken into consideration when planning further studies. 

### Strengths and Limitations

A strength of this study was the study design and the plasma measurements of marine *n*-3 PUFA for evaluation of adherence to the study drug. Another strength was the study duration, which was longer than most studies investigating the effect of marine *n*-3 PUFA on HRV. A limitation was the relatively small sample size, which limited the possibility of further subgroup analysis. Furthermore, this study only analyzed short-term HRV. Measurements of 24 h HRV include nighttime HRV, which is preferable for assessing marine *n*-3 PUFAs’ effect on vagal tone. Almost half of the study population were on antiarrhythmic drugs and, although these medications were not taken before HRV measurements, we cannot rule out that they may have affected the results.

## 5. Conclusions

In renal transplant recipients, *n*-3 PUFA supplementation significantly improved frequency-domain HRV in response to standing, but did not improve time-domain HRV or rHR. As our results were contradictory, additional intervention studies would be needed to clarify the effects of marine *n*-3 PUFA on HRV and possible impact on cardiovascular risk in renal transplant recipients. Such studies should be larger with a longer duration, and include hard-end-points and patients with a low baseline intake of marine *n*-3 PUFA. 

## Figures and Tables

**Table 1 nutrients-11-02847-t001:** Baseline characteristics.

	Marine *n*-3	Control
*N*	66	66
Age, years	52.8 (13.5)	54 (14.2)
Male, %	71	77
Height, cm	175.4 (10.7)	175.5 (9.4)
Weight, kg	79.3 (14.9)	81.1 (14.9)
Systolic blood pressure, mmHg	132.2 (13.8)	135.6 (16.6)
Diastolic blood pressure, mmHg	81.4 (10.6)	82.4 (9.1)
Comorbidity, %		
Hypertension	78.8	63.3
Diabetes mellitus	13.6	19.7
Coronary disease	12.1	12.1
Antiarrhythmic drug, %		
Beta-blocker	48.4	34.8
Verapamil	0.0	1.5
Amiodarone	0.0	1.5
ESRD treatment, %		
Hemodialysis	40.9	50.0
Peritoneal dialysis	28.8	18.2
Preemptive transplantation	29.3	31.8
Time in renal replacement therapy, months	7 (0–19)	9 (0–22)
Living donor, %	21.2	27.3
Marine *n*-3 PUFA, wt. %		
Baseline	6.4 (2.2)	6.3 (2.1)

Data are presented as mean (SD), median (25;75 percentile), or proportion (number of patients) as appropriate. Abbreviations: ESRD = end stage renal disease.

**Table 2 nutrients-11-02847-t002:** Time and frequency domain heart rate variability (HRV) at baseline and at end of study.

	Marine *n*-3 (*N* = 66)		Control (*N* = 66)	
	Baseline	Follow Up	Baseline	Follow Up
TIME DOMAIN				
Resting heart rate, bpm	74.4 (12.6)	71.3 (14.3)	73.0 (10.6)	72.2 (12.0)
SDNN, ms^2^	31.6 (22.6)	39.2 (30.1)	34.0 (24.3)	37.4 (25.3)
Orthostatic ratio	1.1 (1.0–1.1)	1.1 (1.0–1.1)	1.1 (1.0–1.2)	1.1 (1.0–1.2)
E:I ratio	1.1 (1.1–1.3)	1.1 (1.1–1.2)	1.2 (1.1–1.3)	1.2 (1.1–1.4)
Valsalva ratio	1.4 (1.3–1.7)	1.4 (1.2–1.6)	1.4 (1.3–1.6)	1.52 (1.3–1.8)
FREQUENCY DOMAIN				
Orthostatic, ms^2^				
LF	14.6 (4.3–35.6)	20.9 (8.7–62.0)	28.1 * (10.0–99.0)	27.5 (8.0–86.0)
HF	3.0 (0.9–9.0)	6 (0.9–21.6)	7.1 * (1.9–25.1)	6.7 (3.6–18.7)
E:I, ms^2^				
LF	406.5 (165.5–1259.2)	437.0 (132.2–1302.4)	845.5 (133.3–1915.0)	962.8 (233.6–2018.3)
HF	82.0 (13.5–392.6)	70.3 (22.4–403.4)	125.2 (24.9–428.0)	153.1 (36.7–410.0)
Valsalva, ms^2^				
LF	196 (74.4–1063.1)	302.8 (119.9–1063.1)	367.2 (59.0–1192.0)	289.4 (107.4–1743.0)
HF	27.9 (8.5–114.9)	50.3 (20.7–150.3)	50.5 (15.1–158.3)	62.8 (11.5–262.0)

Data are presented as mean (SD) or median (25–75 percentile) as appropriate. Abbreviations: bpm = beats per minute. SDNN = standard deviation of normal-to-normal intervals. E:I = expiratory:inspiratory. LF = low frequency. HF = high frequency. * Significant difference (*p* < 0.05) from marine *n*-3 PUFA.

**Table 3 nutrients-11-02847-t003:** Intervention effect estimates (95% confidence interval) for the ITT and PP population.

	*N*	Intervention Effect Estimate	95% Confidence Interval	*p* Value
TIME DOMAIN				
Resting heart rate (bpm)				
ITT	132	−2.7	(−7.5;2.1)	0.28
PP	102	−3.4	(−8.7;1.2)	0.22
SDNN (ms)				
ITT	132	1.9	(−7.3;11.1)	0.69
PP	102	−1.8	(−12.0;8.4)	0.73
Orthostatic (ratio)				
ITT	132	0.1	(−0.02;0.2)	0.1
PP	102	0.2	(0.01;0.4)	0.04
E:I (ratio)				
ITT	132	−0.01	(0.06;0.05)	0.86
PP	102	−0.01	(−0.07;0.06)	0.85
Valsalva (ratio)				
ITT	132	−0.1	(−0.2;−0.01)	0.04
PP	102	−0.05	(−0.2;0.07)	0.41
FREQUENCY DOMAIN				
Orthostatic test (ms^2^)				
HF				
ITT	132	2.9	(1.1;8.0)	0.04
PP	102	3.7	(1.2;11.2)	0.02
LF				
ITT	132	2.7	(1.1;6.5)	0.04
PP	102	3.3	(1.1;9.6)	0.03
E:I test (ms^2^)				
HF				
ITT	132	0.7	(0.2;2.1)	0.53
PP	102	0.5	(0.2;2.0)	0.38
LF				
ITT	132	0.5	(0.2;1.2)	0.12
PP	102	0.4	(0.2;1.2)	0.13
Valsalva test (ms^2^)				
HF				
ITT	132	1.1	(0.5;2.5)	0.85
PP	102	0.9	(0.4;2.2)	0.82
LF				
ITT	132	1.2	(0.5;2.6)	0.7
PP	102	1.1	(0.4;2.6)	0.87

Abbreviations: ITT = intention-to-treat population. PP = per-protocol population.

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
