# Peer review of "The Effect of Marine *n*-3 Polyunsaturated Fatty Acids on Heart Rate Variability in Renal Transplant Recipients: A Randomized Controlled Trial"

_nutrients, 2019, doi:10.3390/nu11122847_

Round 1

Reviewer 1 Report

The authors have reported here the results of a secondary outcome of an important clinical trial on the effects of 44-wk LC n-3 PUFA supplementation on heart rate variability in patients who have recently undergone kidney transplantation. Due to the high risk of SCD associated with end stage renal disease, and the importance of DHA in neural function and myocyte membrane signalling, this research is timely and the data generated is novel.

Major comments:

Study Design (last paragraph): The primary endpoint is given as change in HRV. It should be made clear here that the primary endpoint of the study was glomerular filtration rate and that the power calculation required 132 patients to be enrolled based on this outcome. Therefore, it should be stated the current paper reports on a secondary outcome of the trial – namely a range of parameters of cardiac autonomic function, known as HRV. Please specify here which HRV parameters were analysed and justify their inclusion (as opposed to the large number of other HRV parameters that were not reported). i.e. if SDNN was the primary parameter of HRV, why this one and not any of the others? Study Design/Measurements: Please explain how the cut-off of 6% was decided upon for low plasma EPA+DHA concentrations and how reliable this marker is for levels of dietary intake, in the general population and in kidney transplant patients. Did you also measure DPA and why was this not included in the plasma measure? Measurements: It would be useful to have a fuller explanation of the purpose of conducting the 3 different cardiovascular reflex tests. What do these tell us about cardiac autonomic function that cannot be obtained from measuring HRV in the resting state, how do they differ, and how is this relevant to risk of arrhythmia? Do they mainly indicate lack of parasympathetic tone or sympathetic overactivity? Is there any evidence that they would be sensitive to small improvements induced by dietary supplementation intervention or are they mainly used as a diagnostic tool for cardiovascular autonomic neuropathy? The reference given on diabetic neuropathy summarises these points but it would be useful for the reader who isn’t familiar with the area to understand the relevance of these tests without having to read the whole reference. Likewise, SDNN, LF and HF were chosen. Why did the authors not also include a more robust time domain parameter of parasympathetic activity such as RMSSD or pNN50? Statistical analysis was conducted by GLM without adjustment for potentially confounding factors – please justify. Median data were used to impute missing baseline data. Which medians were used? Median baselines across the groups? Please defend this decision to impute rather than exclude from analysis. Discussion: The fact that LF and HF were more improved in the n-3 PUFA group relative to the control group in the orthostatic test, but that no differences were found for the other tests, nor for SDNN in any test, needs a bit more unpicking in the discussion. The third paragraph doesn’t fully explore this finding in my opinion – the link with cardiac autonomic response to changes in blood pressure is mentioned, but why was there no corresponding increase in SDNN. Furthermore, it would be helpful if the authors could offer their thoughts on why specifically HF wasn’t increased by EPA+DHA in the deep breathing test and Valsalva manoeuvre.” Discussion: “An omega-3 index has also been used as a biomarker predicting risk of cardiovascular (CV) death [41]. This index defines low levels as < 4wt% and high levels as > 8wt%, corresponding to high vs a low risk for CV-death.” The omega-3 index is obtained from the EPA+DHA content of erythrocyte membranes (not plasma), thought to be a good indicator of longer-term dietary intake and an indicator of tissue status. However, RBC EPA+DHA may not be a good marker in patients with renal failure due to reduced RBC lifespan, nor is it likely that is would be a useful marker in patients who have recently had a kidney transplant. These points should be acknowledged and discussed. The conclusion is very brief and merely restates that main finding without relating it to potential patient benefit. It would be strengthened if there was a statement on how the results might be relevant to clinical practice, and what further research is needed if these results are not considered to be sufficient to change or influence clinical practice.

Minor comments:

Please cite clinical trial registration in methods section I found it a bit difficult to integrate the HRV values for the ITT population in Table 2 with the intervention effect estimates for the ITT and PP populations in Table 3. I suggest that this would be easier to interpret if it was all included in the same table, with the baseline and endpoint values given for both ITT and PP, and the intervention effects for each given in the same row.

Author Response

Point to point response

We appreciate the valuable comments from the reviewers and we thank you for the opportunity to submit a revised version of the manuscript “The effect of marine n-3 polyunsaturated fatty acids on heart rate variability in renal transplant recipients; a randomized controlled trial”.

Reviewer #1:
This paper describes a sub-study of The Effects of Omega-3 Fatty Acids in Renal
Transplantation (ORENTRA) trial to investigate the effects of marine omega-3 fatty acids on heart rate variability (HRV) and resting heart rate (rHR) in 132 renal transplant patients. The results showed that after 44 weeks of three 1-g capsules/d omega-3 fatty acids (Omacor, i.e., EPA + DHA ethyl esters) starting 8 weeks after transplantation resulted in improved frequency domain HRV in response to standing but did not improve time domain HRV nor rHR. There have been no large intervention studies to examine cardiovascular endpoints in renal transplant recipients, despite the fact that sudden cardiac death is a major cause of death in patients with chronic kidney disease. Thus, I think that publication of these results is important, and because it examined an omega-3 fatty acid intervention, I believe that the paper will be of interest to the readership of Nutrients. The paper is generally well written. I have made some specific suggestions for edits below.

Comments:

Please check the journal guidance with regard to spelling out abbreviations at their first use. Also, I believe it is the journal’s preference to use periods as decimals, and not commas.

Also, please be consistent with the number of places reported after decimals. For example, for SDNN in Table 2, you have reported values of 31.6, 39.2, 34, and 37.4. If 34 is actually 34.0, I think you should report it that way for consistency. (This comment pertains to several places throughout the paper.)

Reply: Thank you for pointing this out. This has now been corrected throughout the manuscript.

Abstract: The dose of marine n-3 PUFA is written here as 2,6 g. Suggest that you write it the same way you have written it in the body of the paper as three 1 g capsules/d, each containing 460 mg/g EPA and 380 mg/g DHA.

Reply: This has now been corrected in the abstract.

Abstract: There is no p-value associated with the statement in the abstract about rHR reductions. In my first reading I thought that this was a statistically significant difference. I recommend that a p-value be included in that sentence to make it clear to the reader that this difference, while potentially clinically relevant, did not have a p-value <0.05.

Reply: We have now included the p-value in the abstract.

Introduction, 2nd paragraph: The wording of the 1st sentence is a bit awkward.

Reply: We apologize for the unclear sentence. Changes have been done in the manuscript at page 2, line 13-14.

3rd paragraph: suggest spelling out the name of the ORENTRA trial here.

Reply: As suggested, the full name of the ORENTRA-trial is now added to the manuscript at page 2, line 26.

Materials and Methods, Study Design: How was the dose of 3 g/d chosen?

Reply: The dose was chosen according to the primary endpoint which was eGFR. We aimed for an anti-inflammatory effect, in order to target the inflammation and development of fibrosis in the renal graft. To achieve an anti-inflammatory effect of marine n-3 PUFA relatively high doses are necessary. Calder PC Proceedings of the Nutrition Society (2018), 77, 52–72

Materials and Methods, Measurements, 2nd paragraph: Although the description of the heart rate testing procedures is complete, I think it would be helpful to include a bit more “basic” explanation for readers of the paper who may be reading it for their interest in omega-3 fatty acids or renal medicine, but who may not be as familiar with the heart rate testing procedures. For example, what do 30:15 and RR stand for? Also, the abbreviation E:I is used earlier in the paragraph than where it is described as the expiratory:inspiratory ratio.

Reply: We have clarified RR in the manuscript and corrected the introduction of the abbreviation (E:I): page 3, line 14-27. As for orthostatic 30:15, this includes measurement of response in HR after 15 and 30 seconds after changing position, this has been clarified: page 3, line 14-27.

Materials and Methods, Statistical analyzes (should be analyses): Which variables do you mean exactly when you say that the imputation models included “all” baseline variables. A “two-sided p-value were” should be “two-sided p-value was”.

Reply: We have corrected the grammar accordingly. Imputation models included all baseline variables in Table 1.

Results: Suggest that you explain how the 132 subjects were selected for this study out of the 298 who had received a kidney transplant at the hospital.

Reply: Thank you for this comment. A detailed overview is presented in the ORENTA-trial: Eide I et al. Effect of marine n-3 fatty acid supplementation in renal transplantation: a randomized controlled trial. Am J Transplant. 2019;19(3):790-800.

Results, Subgroup analysis: Here is one of the places where I would suggest consistency of decimal places – the value for rHR reduction is reported as 5 +/-2.8 bpm. Should this be 5.0 +/-2.8 bpm?

Reply: Thank you for pointing this out. We have corrected in the manuscript, page 4, line 39.

Table 2: The abbreviation rHR is used in the text of the Results section, but in this table it just says Heart rate. I would suggest writing this variable the same way throughout the paper (this comment pertains to Table 3 too). Also, because the asterisk is only used in the control column, I think the footnote would be clearer if it said “significant difference (p<0.05) from marine n-3” instead of saying “significant difference (p < 0.05) between groups”. Table 3: The p-value for Valsalva (ratio) ITT in the Table is 0.04, but is reported as 0.03 in the text of the Results section.

Reply: We agree and we also apologize for the discrepancy in the p-value between the result section and Table 3. This has now been corrected in the manuscript (page 4, line 31.

Discussion, 2nd paragraph: Suggest inserting “which was” prior to “8 weeks post-transplant” for clarity. In the last sentence of this paragraph, I think it would be clearer to say “improves autonomic function” instead of “improves autonomic dysfunction”; data suggests should be data suggest; instead of “might add further beneficial effect” suggest writing it as “might have a further beneficial effect.” Discussion, 6th paragraph: I would suggest starting a new paragraph where you say “Finally, previous data” and combining this with the next paragraph that begins “We performed an additional subgroup analysis”. Discussion, 8th paragraph: It is not clear where you say “results from this study” whether you are referring to the present study or to the cross-sectional study in the prior sentence.

Reply: Thank you for your suggestions. We have implemented your suggestions in the manuscript.

Discussion, 11th paragraph: I appreciate that the Discussion section is quite up to date by having included the recently published meta-analysis by Hu et al. in J Am Heart Assoc. You might consider Maki KC, et al. J Clin Lipidol. 2017;11:1152-1160 as another meta-analysis that examined the effect of supplemental long-chain omega-3 fatty acids on cardiac death.

Reply: Thank you for pointing this out. We have added the suggested reference (page 10, line 45).

Strengths and limitations: In the last sentence, “where” should be “were”.

Reply: This has been corrected in the manuscript.

Also, the authors state “Almost half of the study population where on antiarrhythmic drugs, and although these medications were not taken before HRV measurements, we cannot rule out that it may have affected the results.” I believe that the authors should expand on this a bit, particularly the fact that the percentage of subjects taking a beta-adrenergic blocking agent was numerically higher in the intervention group (48.4 vs. 34.8%). Were sensitivity analyses conducted to evaluate whether there was heterogeneity of effect between subjects with and without use of beta-blockers and other antiarrhythmic agents? It does not appear that this was done. However, such an analysis would be of interest.

Reply: We have chosen not to perform a sensitivity analysis as this might be problematic, and include bias. This decision was made according to CONSORT guidelines: http://www.consort-statement.org/checklists/view/32--consort-2010/114-ancillary-analyses

Reviewer 2 Report

This paper describes a sub-study of The Effects of Omega-3 Fatty Acids in Renal Transplantation (ORENTRA) trial to investigate the effects of marine omega-3 fatty acids on heart rate variability (HRV) and resting heart rate (rHR) in 132 renal transplant patients. The results showed that after 44 weeks of three 1-g capsules/d omega-3 fatty acids (Omacor, i.e., EPA + DHA ethyl esters) starting 8 weeks after transplantation resulted in improved frequency domain HRV in response to standing but did not improve time domain HRV nor rHR. There have been no large intervention studies to examine cardiovascular endpoints in renal transplant recipients, despite the fact that sudden cardiac death is a major cause of death in patients with chronic kidney disease. Thus, I think that publication of these results is important, and because it examined an omega-3 fatty acid intervention, I believe that the paper will be of interest to the readership of Nutrients. The paper is generally well written. I have made some specific suggestions for edits below.

Specific Comments to Authors:

Please check the journal guidance with regard to spelling out abbreviations at their first use. Also, I believe it is the journal’s preference to use periods as decimals, and not commas. Also, please be consistent with the number of places reported after decimals. For example, for SDNN in Table 2, you have reported values of 31.6, 39.2, 34, and 37.4. If 34 is actually 34.0, I think you should report it that way for consistency. (This comment pertains to several places throughout the paper.) Abstract: The dose of marine n-3 PUFA is written here as 2,6 g. Suggest that you write it the same way you have written it in the body of the paper as three 1 g capsules/d, each containing 460 mg/g EPA and 380 mg/g DHA. Abstract: There is no p-value associated with the statement in the abstract about rHR reductions. In my first reading I thought that this was a statistically significant difference. I recommend that a p-value be included in that sentence to make it clear to the reader that this difference, while potentially clinically relevant, did not have a p-value <0.05. Introduction, 2nd paragraph: The wording of the 1st sentence is a bit awkward. 3rd paragraph: suggest spelling out the name of the ORENTRA trial here. Materials and Methods, Study Design: How was the dose of 3 g/d chosen? Materials and Methods, Measurements, 2nd paragraph: Although the description of the heart rate testing procedures is complete, I think it would be helpful to include a bit more “basic” explanation for readers of the paper who may be reading it for their interest in omega-3 fatty acids or renal medicine, but who may not be as familiar with the heart rate testing procedures. For example, what do 30:15 and RR stand for? Also, the abbreviation E:I is used earlier in the paragraph than where it is described as the expiratory:inspiratory ratio. Materials and Methods, Statistical analyzes (should be analyses): Which variables do you mean exactly when you say that the imputation models included “all” baseline variables. A “two-sided p-value were” should be “two-sided p-value was”. Results: Suggest that you explain how the 132 subjects were selected for this study out of the 298 who had received a kidney transplant at the hospital. Results, Subgroup analysis: Here is one of the places where I would suggest consistency of decimal places – the value for rHR reduction is reported as 5 +/-2.8 bpm. Should this be 5.0 +/-2.8 bpm? Table 2: The abbreviation rHR is used in the text of the Results section, but in this table it just says Heart rate. I would suggest writing this variable the same way throughout the paper (this comment pertains to Table 3 too). Also, because the asterisk is only used in the control column, I think the footnote would be clearer if it said “significant difference (p<0.05) from marine n-3” instead of saying “significant difference (p < 0.05) between groups”. Table 3: The p-value for Valsalva (ratio) ITT in the Table is 0.04, but is reported as 0.03 in the text of the Results section. Discussion, 2nd paragraph: Suggest inserting “which was” prior to “8 weeks post-transplant” for clarity. In the last sentence of this paragraph, I think it would be clearer to say “improves autonomic function” instead of “improves autonomic dysfunction”; data suggests should be data suggest; instead of “might add further beneficial effect” suggest writing it as “might have a further beneficial effect.” Discussion, 6th paragraph: I would suggest starting a new paragraph where you say “Finally, previous data” and combining this with the next paragraph that begins “We performed an additional subgroup analysis”. Discussion, 8th paragraph: It is not clear where you say “results from this study” whether you are referring to the present study or to the cross-sectional study in the prior sentence. Discussion, 11th paragraph: I appreciate that the Discussion section is quite up to date by having included the recently published meta-analysis by Hu et al. in J Am Heart Assoc. You might consider Maki KC, et al. J Clin Lipidol. 2017;11:1152-1160 as another meta-analysis that examined the effect of supplemental long-chain omega-3 fatty acids on cardiac death. Strengths and limitations: In the last sentence, “where” should be “were”. Also, the authors state “Almost half of the study population where on antiarrhythmic drugs, and although these medications were not taken before HRV measurements, we cannot rule out that it may have affected the results.” I believe that the authors should expand on this a bit, particularly the fact that the percentage of subjects taking a beta-adrenergic blocking agent was numerically higher in the intervention group (48.4 vs. 34.8%). Were sensitivity analyses conducted to evaluate whether there was heterogeneity of effect between subjects with and without use of beta-blockers and other antiarrhythmic agents? It does not appear that this was done. However, such an analysis would be of interest.

Author Response

Reviewer #2:
The authors have reported here the results of a secondary outcome of an important clinical trial on the effects of 44-wk LC n-3 PUFA supplementation on heart rate variability in patients who have recently undergone kidney transplantation. Due to the high risk of SCD associated with end stage renal disease, and the importance of DHA in neural function and myocyte membrane signaling, this research is timely and the data generated is novel.

Comments:

Study Design (last paragraph): The primary endpoint is given as change in HRV. It should be made clear here that the primary endpoint of the study was glomerular filtration rate and that the power calculation required 132 patients to be enrolled based on this outcome. Therefore, it should be stated the current paper reports on a secondary outcome of the trial – namely a range of parameters of cardiac autonomic function, known as HRV.

Reply: Thank you for pointing this out. We agree and have clarified this in the manuscript (page 2, line 46-48).

Please specify here which HRV parameters were analysed and justify their inclusion (as opposed to the large number of other HRV parameters that were not reported). i.e. if SDNN was the primary parameter of HRV, why this one and not any of the others?

Reply: The following HRV matrices were used: In the time domain we used Standard Deviation of Normal-to-Normal intervals (SDNN). The HRV parameter SDNN is a measure of combined sympathetic and parasympathetic activity. In the frequency domain we used the low frequency (LF) 0.04-0.15 Hz, and the high frequency 0.15-0.4 Hz components. The LF components is influenced by sympathetic, parasympathetic and baroreflex sensitivity. The high frequency (HF) band from 0.15-0.4 Hz is influenced by parasympathetic and the normal breathing rhythm mainly contributing to power with a center frequency around 0.3 Hz.

Study Design/Measurements: Please explain how the cut-off of 6% was decided upon for low plasma EPA+DHA concentrations and how reliable this marker is for levels of dietary intake, in the general population and in kidney transplant patients.

Reply: We thank the reviewer for this question. The cut off of 6% was chosen based on the ORENTRA-trial where 6% wt was the median in both the intervention and control group at baseline. Plasma levels of marine n-3 PUFA might be a better marker of dietary intake than dietary interviews alone, due to reporting bias. However, we have previously shown that reported dietary intake of marine n-3 PUFA corresponds to measured levels in patients with kidney disease. Svensson M et al. The effect of n-3 fatty acids on plasma lipids and lipoproteins and blood pressure in patients with CRF. Am J Kidney Dis. 2004;44(1):77-83. Svensson M et al. The effect of n-3 fatty acids on lipids and lipoproteins in patients treated with chronic haemodialysis: a randomized placebo-controlled intervention study. Nephrol Dial Transplant. 2008;23(9):2918-24.

Did you also measure DPA and why was this not included in the plasma measure?

Reply: We have measurements (wt%) of all fatty acids with gas chromatography. However, DPA is only a very small amount of total marine n-3 PUFA: In addition, the intervention contained EPA and DHA and we have not reported DPA in our original publication. Eide I et al. Effect of marine n-3 fatty acid supplementation in renal transplantation: a randomized controlled trial. Am J Transplant. 2019;19(3):790-800.

Measurements: It would be useful to have a fuller explanation of the purpose of conducting the 3 different cardiovascular reflex tests. What do these tell us about cardiac autonomic function that cannot be obtained from measuring HRV in the resting state, how do they differ, and how is this relevant to risk of arrhythmia? Do they mainly indicate lack of parasympathetic tone or sympathetic overactivity? Is there any evidence that they would be sensitive to small improvements induced by dietary supplementation intervention or are they mainly used as a diagnostic tool for cardiovascular autonomic neuropathy? The reference given on diabetic neuropathy summarises these points but it would be useful for the reader who isn’t familiar with the area to understand the relevance of these tests without having to read the whole reference.

Reply: Thank you for these questions. Cardiovascular autonomic reflex tests are easy to perform, sensitive, specific, reproducible and standardized. Therefore, they are considered the gold standard measures of autonomic function. Adverse outcomes in the cardiovascular reflex tests have higher predictive power for cardiovascular events such as MI, stroke and sudden cardiac death than the classical risk factors such as smoking, LDL levels and family history of coronary artery disease. The DIAD study. Diabetes Care (2004) 27:1954–61. doi:10.2337/diacare.27.8. 1954

The deep breathing test and responses to standing are largely a measure of parasympathetic function, whereas the Valsalva maneuver is a measure of both parasympathetic, sympathetic function and baroreceptor function. The DIAD study: a randomized controlled trial. JAMA (2009) 301:1547–55. doi:10.1001/jama.2009.476

Spectral analysis of passive HRV testing is used extensively for research purposes and generally believed to supply information about both sympathetic and parasympathetic modulation earlier than active tests. However, we have shown that active cardiovascular reflex tests are more reliable to detect autonomic dysfunction than passive HRV tests. Journal of Diabetes Science and Technology 2015, Vol. 9(3) 681–686. DOI: 10.1177/1932296814567226

Likewise, SDNN, LF and HF were chosen. Why did the authors not also include a more robust time domain parameter of parasympathetic activity such as RMSSD or pNN50?

Reply: Detailed analysis, including both cardiovascular reflex tests and frequency analysis, are an important tool to estimate cardiac autonomic nervous activity. The cardiovascular reflex, deep breathing and response to standing are robust measures of the parasympathetic activity, and therefore RMSSD nor pNN50 were calculated.

Statistical analysis was conducted by GLM without adjustment for potentially confounding factors – please justify. Median data were used to impute missing baseline data. Which medians were used? Median baselines across the groups? Please defend this decision to impute rather than exclude from analysis.

Reply: We have chosen to impute missing baseline data rather than exclude according to ITT principles. This decision was made according to CONSORT guidelines: http://www.consort-statement.org/checklists/view/32--consort-2010/114-ancillary-analyses.

Discussion: The fact that LF and HF were more improved in the n-3 PUFA group relative to the control group in the orthostatic test, but that no differences were found for the other tests, nor for SDNN in any test, needs a bit more unpicking in the discussion. The third paragraph doesn’t fully explore this finding in my opinion – the link with cardiac autonomic response to changes in blood pressure is mentioned, but why was there no corresponding increase in SDNN.

Reply: We agree. There is only tendency towards that SDNN is more improves in the n-3 PUFA group relative to the control group (at follow up 39.2 ms vs 37.4 ms). Indicating that it may be due to relatively small sample size and/or duration of the study. Also, we cannot rule out that half of the study population were on antiarrhythmic drugs, and although these medications were not taken before HRV measurements, it may have affected the HRV results.

Furthermore, it would be helpful if the authors could offer their thoughts on why specifically HF wasn’t increased by EPA+DHA in the deep breathing test and Valsalva manoeuvre.”

Reply: Thank you for this comment. The HF band from 0.15-0.4 Hz is mainly influenced by normal breathing rhythm, which contributes to power with a center frequency around 0.3 Hz. Deep breathing for 1 minute with a respiratory frequency of 6 breaths/min (experimentally induced variations of RR at 0.1 Hz) will generate large increased LF with a central frequency of 0.1 Hz.

Discussion: “An omega-3 index has also been used as a biomarker predicting risk of cardiovascular (CV) death [41]. This index defines low levels as < 4wt% and high levels as > 8wt%, corresponding to high vs a low risk for CV-death.” The omega-3 index is obtained from the EPA+DHA content of erythrocyte membranes (not plasma), thought to be a good indicator of longer-term dietary intake and an indicator of tissue status. However, RBC EPA+DHA may not be a good marker in patients with renal failure due to reduced RBC lifespan, nor is it likely that is would be a useful marker in patients who have recently had a kidney transplant. These points should be acknowledged and discussed.

Reply: We agree that RBC measurements of EPA + DHA might be problematic in patients with renal failure and a recent transplant. However, we only measured plasma levels in our study and the omega-3 index was included only as a clarification for the reader regarding high and low levels in the discussion.

The conclusion is very brief and merely restates that main finding without relating it to potential patient benefit. It would be strengthened if there was a statement on how the results might be relevant to clinical practice, and what further research is needed if these results are not considered to be sufficient to change or influence clinical practice. 

Reply: Thank you for this comment. We have added a sentence according to your suggestion (page 11, line 15-19).

Please cite clinical trial registration in methods section.

Reply: We have added this in the methods section (page 3, line 42).

I found it a bit difficult to integrate the HRV values for the ITT population in Table 2 with the intervention effect estimates for the ITT and PP populations in Table 3. I suggest that this would be easier to interpret if it was all included in the same table, with the baseline and endpoint values given for both ITT and PP, and the intervention effects for each given in the same row. 

Reply: We have reported data according to CONSORT guidelines which states that for each outcome, study results should be reported as a summary of the outcome in each group, together with the contrast between the groups, known as the effect size. According to that Table 2 are raw data and Table 3 shows effect size. http://www.consort-statement.org/checklists/view/32--consort-2010/111-outcomes-and-estimation